# Guidelines on Designing Conceptual Framework for Strategic Management with Application to the Energy Industry

**Maryam Ayoubi [1], Ehsan Mehrabanfar [2],\* and Audrius Banaitis [3]**

1   Research Institute of Petroleum Industry, Tehran, Iran; aioubi.m@gmail.com
2   Young Researchers and Elites Club, North Tehran Branch, Islamic Azad University, Tehran, Iran
3   Department of Construction Management and Real Estate, Vilnius Gediminas Technical University, Saulėtekio al. 11, 10223 Vilnius, Lithuania; audrius.banaitis@vgtu.lt
\*   Correspondence: e.mehrabanfar@aut.ac.ir

**Abstract:** The role of strategic management is to handle the day-to-day needs of the firms with respect to current events of fluctuating markets so that they can effectively reach to their pre-planned, long-term goals. This study aims to explain the steps taken to design a new conceptual framework in the strategic management department of the National Iranian Gas Export Company (NIGEC), as well as the main components of the conceptual framework. It first goes through the logic behind the techniques applied to design this conceptual framework, and then each component is explained as to why it works as part of an integrated conceptual framework. This new framework is part of a change regime in the organisation along with the whole oil industry in Iran aimed at contributing to directing the organisations confronted with rapidly altering environments. The insights of this study shed light on the basics of designing an adjusted conceptual framework for managers and researchers, and all companies faced with constant internal and external challenges and other characteristics of modern organisations.

**Keywords:** conceptual framework; strategic management; organisational change; oil industry; Iran

## 1. Introduction

The oil industry has been the central axis of Iran's economy by making generous wealth for all years from 1901 until today. It accounts for a large share of current gross domestic product and still stands as the basic core of economic growth. Such a significant role makes the subject of strategic management in the oil industry highly critical, however, the recent changes in the energy market have even inevitably increased the importance of strategic management in all associated firms. The rise of unpredictability in the trends of the markets and altering incentives of multiple players involved are among the most important changes. All firms in the oil industry in Iran are now adapting themselves to the possible future trends of the market, as they have started a transferal movement from traditional management to the implementation of new techniques which can be quite obviously tracked over the last five years. The signs of these attempts can be observed in different projects taken to revise the structure and processes in the whole oil industry. There is not even a firm left that has not faced at least one soft change in the core of its business. For similar cases explaining such changes look at Mortazavi Ravari et al. (2016) and Alizadeh et al. (2016), which both discuss rather complex and advanced techniques used to help a company in the Iranian energy sector improve its internal process. However, the list of similar publications which are taken from industrial projects goes on and there are many papers aligned with the same aim as this study in the literature.

Being one of the main departments of the National Iranian Oil Company, the National Iranian Gas Export Company (NIGEC) has been no exception in this regard. In order to improve the efficiency of the organisation and make sure this firm is congruent with internal and external environments, top managers of NIGEC have decided to redesign the conceptual framework used in the department of strategic management in previous years and replace it with a new framework which is customized to all modifications in the current internal and external circumstances and is also consistent with the current literature of strategic management.

NIGEC, like other national oil companies, is directed by the National Iranian Oil Company (NIOC), which, itself, is directed by Ministry of Petroleum. Accordingly, the structure and prevalent management systems of NIGEC are no different from the established disciplines of the public sector in Iran which is attached to an oversized bureaucracy, slow-decision making, and traditionalism, factors that have arguably caused the firms not to reach their strategic goals in recent years. However, due to being known as a general problem, all main companies under the control of the petroleum ministry have gone through multiple changes. CEOs have, themselves, accepted the necessity of the change and have mandated departments to implement basic organisational changes. NIGEC has initiated to make changes in the strategic management department due to the essence of its work which is naturally challenging and, on the other hand, it also needs stability and long-term planning. Being an exporting company, it needs to deal with international markets on a daily basis, which makes its outcomes quite sensitive. Thereby, it was clarified for top managers of NIGEC that going through a primary change in the strategic management department is an absolute necessity.

How can a firm handle the altering environments and complex markets that vary every day? NIGEC, like any other firm, was already working based on a traditional management system for optimally organizing the interactions between itself and its altering environment. However, when firms are forced to handle complex management issues raised due to rapid changes in their environments, they cannot be easily managed anymore. Strategic management is able to sustain the efficiency of such firms overwhelmed by unknowns of the future since it is basically developed to help managers direct their organizations through complex and uncertain environments. However, for this aim, strategic management firstly needs to be based on an appropriate conceptual framework that is entirely adapted with respect to the essence of the organization since it is the conceptual framework that undeniably acts as the basic core of the strategic management software.

Each organization has its own specific values, vision, and mission, as well as structural and environmental characteristics to which it owes its very unique existence. These are the critical elements to build up a unique conceptual framework that enables the organisation to react to altering environments efficiently. Yet, they are not adequate, because the conceptual framework should be integrated with appropriate techniques and methods taken from the latest literature. The main issue, then, is to devise a framework that is nurtured from the central elements within the organisation and is also supplemented with accordant methods.

The conceptual framework designed in NIGEC is described step-by-step in the following sections. The second section provides a literature review of the primary concepts of the study. The third section, which is also the main chapter of this study is about strategic management procedures. It shows the framework and explains the components of the framework. The fourth section is dedicated to the conclusion.

## 2. Literature Review

### 2.1. Evolution of Strategic Management

The concept of strategy goes back to at least 340 B.C. In his eternal book titled The Art of War, Sun Tzu, who was a Chinese army commander, describes principles of winning his wars by using principles of today's strategic management. He clearly was the main source of many concepts in current strategic management literature. However, the concept of strategy in its modern form is a

post-World War II product necessitated due to the evolutionary process of planning. Before the Second World War, organizations used to provide annual planning or budgeting for estimating and predicting their costs and incomes. However, after World War II, technological advances, the development of communication devices, and air travel introduced greater mobility in business activities making long-term planning a necessity. The concept of long-term planning, through which the total operations of an organization within a three to five-year horizon was planned, is, in fact, a product of this era (Rezaei 2004).

In the early 1960s, the U.S Ministry of Defense decided to formulate the experiences obtained from strategic decision-making during World War II (Gluck et al. 1980). This decision forms the basis of strategic planning as a systematic formulation of organisational activities in a specific period (Bryson 2010). Two years later, Chandler (1962), a professor at Harvard University, introduces these concepts to the business world. In 1965, Andrews (1965) introduces a paper, i.e., the "business strategy" based on Chandler's views and theories. Igor Ansoff, the Director of Lockheed Electronics, welcomes this approach and implements it in business environments (Gluck et al. 1980). Ansoff's successful implementation of the strategic approach attracted attention to the introduced concepts and methodologies. In the early 1970s, the Boston Consulting Group added the product portfolio methodology (BCG matrix) to the strategy literature.

The year 1980 is another milestone since it is the year that strategic management becomes a field in academic studies after that "Pittsburg conference is held to define a paradigm for business policy" (Lyles 1990). It took about two decades for the term of strategic management to be radically separated from strategic planning, which was achieved by the establishment of a specific journal in strategic management in 1980. Michael Porter's book on five forces, entitled *Competitive Strategy*, extends the space further in the literature and establishes the basis of other theories for the next decades (Edwards et al. 2014). Throughout the 1980s, Michel Porter's theories on competitive advantage and competitive scope dominate the scene (O'Shannassy 2001).

From 1980 to 1990, strategic management goes through a maturity process at two basic levels, i.e., the methodologies and the variety of topics addressed, which both diversify gradually. Topics such as global competition, globalization, innovation, and technology receive more attention from scholars in the field during these years. In addition, positivist methodologies are replaced with methods that are adequately aligned with applied research contexts (Lyles 1990). Gradually, methodologies become more quantitative by replacing in-depth studies, and further, a global consensus forms over the definitions of different notions in strategic management (Guerras-Martin et al. 2014). Finally, in the 1990s, the new age of approaches in strategic management, with creativity as their core element of strategic effectiveness, is initialized with the views expressed by reputable scholars such as Henry Mintzberg and Gary Hamel.

In the framework presented by Guerras-Martin et al. (2014), two main criteria i.e., macro-micro and external-internal are introduced to understand how different factors have shaped the format of strategic management in all these years. They include factors such as *transaction cost economics* and *agency theory* in the mainstream of strategic management in the 1980s to the 1990s which are explained to be both partially internal and external, and likewise macro and micro. However, besides these two early factors which are omitted in the next decades, there are other factors in the literature that appear in Guerras-Martin et al.'s framework as to clarifying the evolution of the mainstream, including *resource-based view* (RBV) for after the 1990s which is explained to be as macro-internal, *behavioral strategy for* after 2005 as micro-internal, *institutional approach* after 2002 as external-macro, and *entrepreneurship-based* after 2000 as external-macro. They use the metaphor of *tension* to show the *swing* between internal and external and micro and macro over these years. In the 1960s, which is the birthdate of strategic management, the internal factors are considered more important than the external ones due to the need to focus on strengths and weaknesses among top firms. However, up to the 1970s, gradually both dimensions get considered, which is also reflected in methodologies like SWOT. From the 1970s to 1980s it becomes a matter of external importance in which the notion of

industry structure becomes central. At the end of the 1980s, again, the pendulum reaches a middle position, and in the next years, the internal factors once more lead the pendulum, while of course, the significance of external factors still partially matters.

For the macro-micro pendulum, Guerras-Martin et al. (2014) argue both dimensions existed somewhat equally in the literature, however, later in the 1980s the macro dimension attains a partial lead. RVB in the 1990s even pushes this trend, but in the middle of the 1990s, knowledge-based views change the position in favour of micro to shed light on how firms, in reality, can develop competitive advantages. In recent years, the notion of behaviour strategy has pushed the micro level as well. In a nutshell, in the last five decades, strategic management has gone from an *external-macro lead* to an *internal micro-lead*. However, this configuration does not imply that the impact of the other side of the pendulum is no more considered in the actuality of the firm's business, it just describes the trends in the mainstream of strategic management literature. Overall, this framework is a basic overview of the literature to show the path taken to reach the current position.

### 2.2. Basics of Strategic Management

The word "strategy" is derived from the Greek term *strat gous* or *strat geiy*, meaning the art of generals, since, in the army, the strategy is a responsibility for generals, just as it is similarly a responsibility for managers in the business. Different definitions have been proposed for strategy, while each focuses on a specific dimension. Bruce Henderson, the eminent scholar and founder of the Boston Consulting Group, defines strategy as, "creating a unique advantage that differentiates an organization from its competitors" (Kiechel 2010). He believes that the main task is to "manage this distinction". In another definition: "strategy is a comprehensive program for an action that determines the general orientation of an organization and provides resource allocation guidelines for achieving long-term organizational goals" (Ansoff 1997). It can be argued, however, that "strategies are tools through which companies can achieve their long-term goals"; additionally, it can be said that "strategy is the comprehensive and parent plan of a company for determining how to achieve its mission and goals". Strategy maximizes competitive advantage and minimizes competitive shortages (David 2001).

According to Mintzberg, the strategy is a product of different views and opinions developed in accordance with various schools of strategy. Meanwhile, some of these definitions may differ from the traditional definitions. It should be mentioned that, through the years, the term of "strategy" has been defined, used, and interpreted in different ways based on different management or strategy theories (Mintzberg et al. 1998; Mintzberg and Lampel 1999).

In terms of significance and scope, strategies can have three levels, i.e., corporate, business, and operational. Clearly, corporate strategies are more significant as they involve the whole firm and are a basis for other units. However, business strategies just concern one specific sector, and operational strategies remain at the level of operation (Johnson et al. 2016).

### 2.3. Steps of Strategic Management

The first step in strategic management is formulation. The main aims of this step are to determine the company's mission, vision, and long-term goals, and then identify external threats and opportunities, as well as internal weaknesses and strengths. Since no organization has unlimited resources, strategists should choose the one among different strategies that end in the most beneficial scenario. Nevertheless, strategic decisions, whether right or wrong, have multi-dimensional impacts and long-lasting effects on organizations (Aliahmadi et al. 2004).

Strategy implementation is the second step and requires considering annual goals, determining policies, and allocating resources in a manner that the formulated strategies can be actualized. Strategy implementation demands development of a congruent culture to support the adopted strategies, the foundation of an effective organizational structure, directing marketing efforts, budgeting, creating, and employing information systems, and compensating employees for their services. Strategy implementation means that employees and managers should be mobilized towards putting the

formulated strategies into practice (Hill et al. 2014). It is generally assumed that strategic management, in its implementation phase, is the most difficult step and necessitates employees to be committed to their organization, to make sacrifices, and to exert to some extent self-control (Gamble et al. 2014).

The success of this phase depends on the ability of managers to motivate their employees, which, indeed, is more an art than science because strategies that are merely formulated, but not implemented, are nothing but a waste of time. Managers should have distinguishing skills in making personal relationships in order to successfully implement the formulated strategies. The main factor is to encourage all employees to take part in, and devote themselves to, their respective tasks. They should do their best for achieving the determined organizational goals (Aliahmadi et al. 2004).

In the process of strategic management, evaluation of strategy is the final step. Managers need to know the circumstances where their strategies will not work. Basically, strategy evaluation implies that relevant data should be collected as well. All strategies will be changed in the future due to the nature of internal and external factors. Strategy evaluation should be practised nonetheless because there is no guarantee for today's success to extend to even the near future. An organization that relies only on its current state will soon become arrogant and eliminated (David 2001).

### 2.4. The Importance of Change in Strategic Management

Usually, organisations lag behind the changes in their environment which end in a "strategic drift". This deviation in the strategic management occurs when the firm is not responsive enough to the altering environments as it should be (Johnson et al. 2016). Even though there might have been "incremental fine-tuning adaptation" (Hayes 2014), but there are minor changes which are not able to make a course of direction for the strategic management. Strategic drift can even lead the processes of strategic management to a state of "flux" where strategies change, but not in a specific and clear direction (Johnson et al. 2016).

Nevertheless, the matter of the pace of change in strategic management is rather ambiguous since, in the eyes of the managers, strategies are always changing. When faced with a new problem, managers design and implement a new solution. Most of the time they cannot get the results, and just in a handful of times, they end in success. However, the ambiguity is not in the existence, but in the pace, of change (Hayes 2014). The environment of the organisations essentially does not obey the same pace of changes in the organisation. Not only should the organisation be a place of continuous change, but also aligned with a responsive pace to the environment. This matter is usually neglected. If the change is too fast in the external side, then the managers suddenly might need to implement a transformational change in the inside which is exacting and risky; or, they might avoid such change until it is too late which is even worse (Johnson et al. 2016). The reason why the change might not happen at the right time is discussed abundantly through the lens of "cultural and behavioural selection contingencies", which are considered to be the roots of incapability of the organisation to react promptly (Glenn and Mallot 2004).

### 2.5. Conceptual Framework for Strategic Management

The literature on the topic of conceptual frameworks used in real practice in the firms across different industries is limited, which is due to the importance of information in the competition that exists among all firms. Thereby rather than face many specific or detailed conceptual frameworks in the literature, the purpose and the techniques used in designing the conceptual framework are explained. For instance, Woods et al. (1998) explain the role that a conceptual framework can play in the strategic management of the subsector of the agriculture industry as it can lead to the enhancement of performance when the firms are conceptually aligned without, but it does not provide any further elaboration on any specific framework. Mazzarol (2005) focuses on the specifics of the conceptual framework of small firms and outlines questions that can help managers of these firms to design their framework correctly, which is to balance strategy with resource and structure. Mazzarol and Reboud (2006) investigate this theoretical framework in a case study in which they emphasise on the significance of balance between

strategic thinking and operational management. Knott and Thnarudee (2008) discuss the specifics of the conceptual framework in a multi-unit firm's corporate strategy and emphasise the importance of coordination and communication among different units.

Sahoo et al. (2011) propose a conceptual framework for strategic management technology in the automobile industry in India. They emphasise on the importance of conceptual frameworks for decision-makers and the paths they choose for the future of their organisation. They claim there are very few practical conceptual frameworks available in the literature and, thereby, they design a simple framework for the automobile industry and technology management in which they link technology strategy with business performance through technology capability. To develop this framework, they take a step-by-step process from studying the literature and obtaining expert opinions to the identification of the factors for strategic technology management and then interpretative equation modelling. In another similar study, Sahlman and Haapasalo (2012) discuss the elements of the strategic management of technology. They argue the most complex tasks of the firm are to develop well-structured definitions for the elements of the conceptual framework and then use them to develop their own conceptual frameworks.

## 3. Development of the Framework

### 3.1. The Methodology

The methodology used in this study is a combination of a literature review, a panel of experts, and fieldwork. Expert panels in this study play the most important aspect of the methodology since the main aim is to match the framework for a specific company, which is already occupied by a strategic management department, thereby the judgements of experts panel towards the reality of the practice of the framework in the organisation, when it is needed, largely plays the determinative part.

Nonetheless, the literature is the starting point to design the conceptual framework. The importance of organisational learning and the role that interaction with strategic change plays in the successful implementation of the strategy is abundantly discussed, but how it is possible to make an alignment between unknown change and organisational learning? The proposed steps to achieve this objective are based on the following principles:

1.  The paradigm of the framework should be systemic and basically interactive with the organisation and, by any means, it should not be predetermined.
2.  The foundation of carefully knowing the environments, as well as accepting change as a necessary part of the organisation, should be adopted across all sectors of the organisation.

Based on the framework of Mintzberg, there are two paradigms in strategic management. The proposed framework in this study is a systemic approach to strategic management which is arguably more fitting with the requirements of the modern organisations, such as NIGEC (Gregory 2007). This framework is a combination of both descriptive and prescriptive paradigms and does not belong to either of them as it depicts an interactive approach to the ideal future. By this means, this framework is, intentionally, a systemic approach to strategic management that combines different techniques as required. Unlike other ones, this framework starts the strategic formulation by defining and determining the ideal point i.e., *the future to be*. The product of this approach is an ideal statement of the vision and mission which, together, determine the ideal future and the ideal state of the organization regardless of the existing limitations. By moving from the ideal state towards the current state, this framework exploits the maximum capability of organizational learning.

A modern perspective in strategic management methodology called "value-focused thinking" (VFT) is then used to develop the *basic goals hierarchy* as well as the *final cross-sectional goals network*, and also to identify the basic and cross-sectional goals. According to the VFT logic, values should be considered during decision-making to develop better options. Each basic goal should be categorized into other sets of quantitative and sectional goals. This process needs the help of the panel of experts.

Afterwards, basic goals are structured within a hierarchy framework while sectional goals are shown in the network. In the network of sectional goals, it is shown which one affects the other sectional goals. All the sectional goals eventually influence the basic goals which are depicted in the *network goals* (León 1999; Kotler and Armstrong 1998; Alencar et al. 2017).

In the hierarchy of basic goals, a low ranked goal is a part of a high ranked one. In other words, a high-rank goal is directly defined by the set of the goals ranked in lower levels in the hierarchy. At least two lower-ranked goals should be related to each higher-ranked goal. The hierarchy of goals is defined in the following three levels:

- Basic goals, which are basically important.
- Middle goals, which just meet the basic goals.
- Functional goals, which meet middle goals and are measurable.

After identifying the network goals, other factors, including strategic concepts and strategic plans, are agreed based on the judgment of the panel of experts. These factors are then analysed for identification of their stakeholders and environmental factors as well as the key criteria for success. Finally, analysis of internal and external environment is conducted and, afterwards, scenario planning is implemented. Obviously, to understand the framework provided in Figure 1, the flow of the process should be started from the last column, which is also true about Figure 2. However, in Figure 1, the two blue charts are the final steps.

Overall, the method used in this framework is based on a practical tool in future studies, i.e., backcasting. This tool is used when the trends are not clear and the environments are complex (Holmberg and Robert 2000). The use of backcasting for designing the conceptual framework of NIGEC is consistent with the needs and circumstances of the organisation.

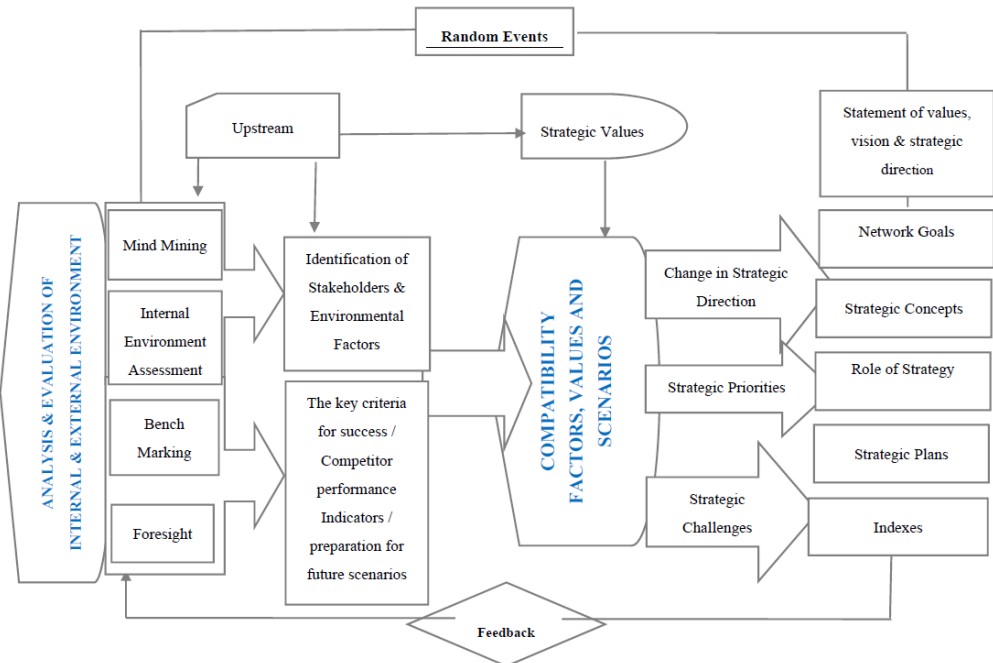

**Figure 1.** The conceptual framework designed for strategic management in NIGEC.

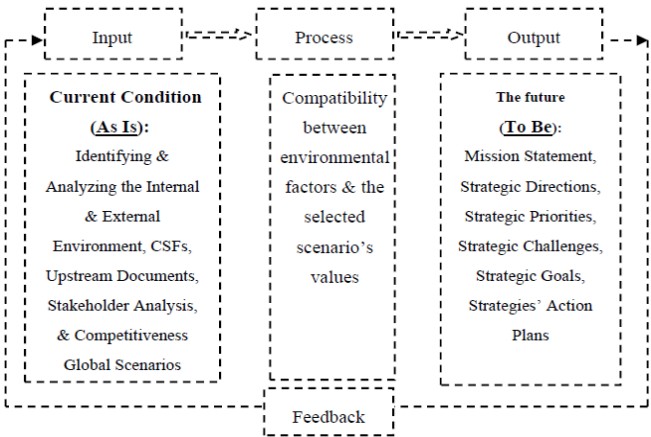

**Figure 2.** View of the strategic framework from input to output, i.e., from now to the future.

### 3.2. Obscure Components of the Framework: Preventing Bottlenecks and Competency

New opportunities need to be discovered for their potential benefits. However, finding these opportunities is not simple while it directly necessitates the existence of specific factors in the organisation linked with the potentials in the framework. The conceptual framework should be designed so that it can deal with all types of possible benefits and opportunities.

The first group of opportunities includes benefits that are not complex and can be found by the firm. They are not important in the strategic management area, but they should be considered in the planning phase in order to prevent them from turning into bottlenecks for the firm. The second group of benefits includes opportunities which are rather difficult to find but are currently available in the organization. These are the *core competencies* of the organization and play a vital role in creating the competitive advantage for the organisation. Core competencies are the strengths of an organisation in dealing with issues which are bottlenecks for other organizations. The role of the conceptual framework for this type is to identify and work on their progress. The most important group of the opportunities is the second type since they certify the existence of the firm in the current position in the market. The third group of benefits can be achieved with extreme difficulty. These opportunities are called critical success factors (CSFs), or bottlenecks (Kotler and Armstrong 1998).

At this stage, the organisation turns to have a unique competency and is enabled to be the leader of the market. The role of the conceptual framework towards this type is rather unclear since it is a bottleneck that is unknown for the organisation, however, dealing with the issues and technology management might lead the firm to develop such opportunities, and then create a new competency. The conceptual framework should be able to respond to the confrontation with unknown bottlenecks for this purpose. Understating of the concept of bottleneck and opportunities is critical for the whole team who deals with the conceptual framework. This explanation is mentioned here to shed light on an obscure part of the conceptual framework which is not directly reflected in components of the framework but is required to be explained. This unseen part is, rather, a checkpoint, since if the framework cannot provide the ability to cover all types of bottlenecks then it needs to be redeemed.

### 3.3. Components of the Framework: Analysis & Assessment of Internal and External Environments

First of all, determination of the organizational environment depends on an explicit definition of the organization and its boundaries. Organizations may have different environments depending on factors such as size, type of activity, the scope of activity, and so on. Experienced managers should be aware of all these factors in order to effectively analyse them. Generally, an environment is classified into two different categories: the internal environment and the external environment.

The internal environment covers those factors and all components which are internally connected to the organization, such as mission, goals, and structure. It regards how to handle challenges and use

opportunities, in addition, to measure the performance of main functional and operational strategies in retaining current customers and attracting potential customers in a competitive environment, as well as make financial statements and perform liquidity monitoring. It covers subjects like efficiency, efficacy, effectiveness, technological power, organizing quality, philosophy of values, plans, and functional measures for doing routine organizational tasks.

On the other hand, the external environment includes all external conditions, flows, and factors which potentially influence the organization, but cannot be controlled by the organization (Daft 2007). In other words, every external component that is associated with a sub-system of the organization is a part of the whole external environment. For example, technological changes in the environment are directly related to R and D and production departments of the organisation and, thus, an element of the external environment (Amirkabiri 2003).

To conduct a comprehensive study on the environment and its related factors, this study divides the external environment into the international and the functional, and the internal environment into the macro (national) and domestic industries. Figure 3 shows the level of internal and external boundaries of the environment.

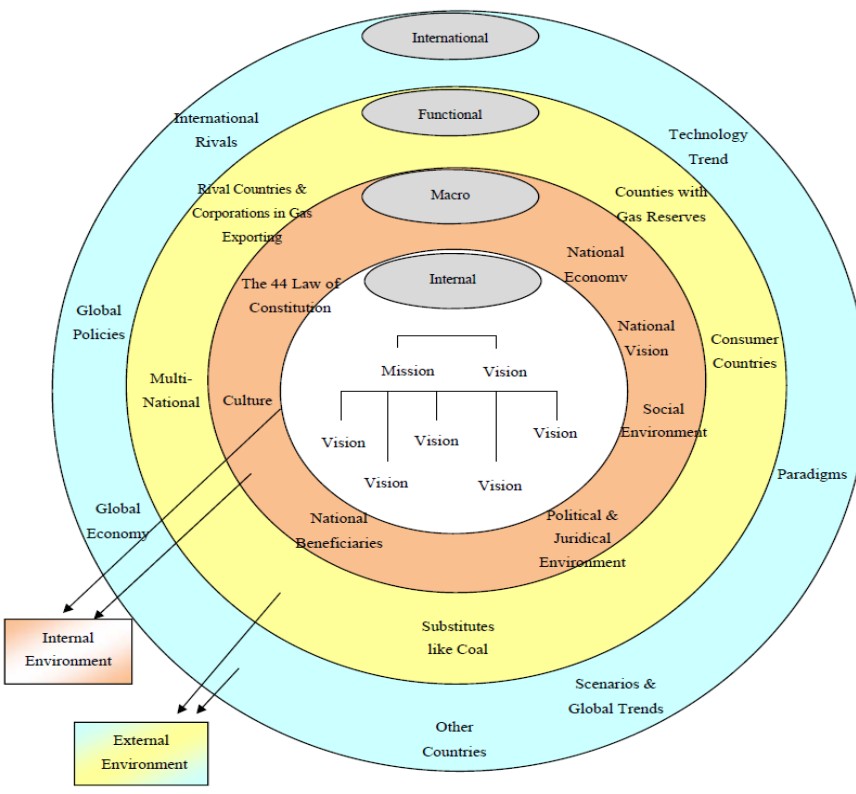

**Figure 3.** The internal (macro, domestic) and external (international, functional) environments of NIGEC.

### 3.4. Components of the Framework: Compatibility Factors, Values, and Scenarios

In the framework, the probable scenarios of the industry should be studied via foresight methodologies. This study helps to develop scenarios and select the best fitting one for the firm. In the process of analysing the external environment, upstream documents (e.g., constitutions such as 44 law which mandates general policies of privatization in Iran, official and legal national documents, etc.) are of vital importance. These documents shape the fundamental trends that impact on the industry, as well as the strategic values of the firm. By analysing the competitiveness and stakeholders and, combining them with the extracted scenarios, as well as the strategic factors obtained from

analysis of external and internal environments, the framework reaches the compatibility factors, values, and scenarios.

## 3.5. Making Strategic Decisions and Developing Strategies

In strategic management, after completion of evaluating external and internal environments and determining important weaknesses, strengths, threats, and opportunities, the process of making strategic decisions starts (Hinz 2009). Figure 4 shows the steps taken to make decisions.

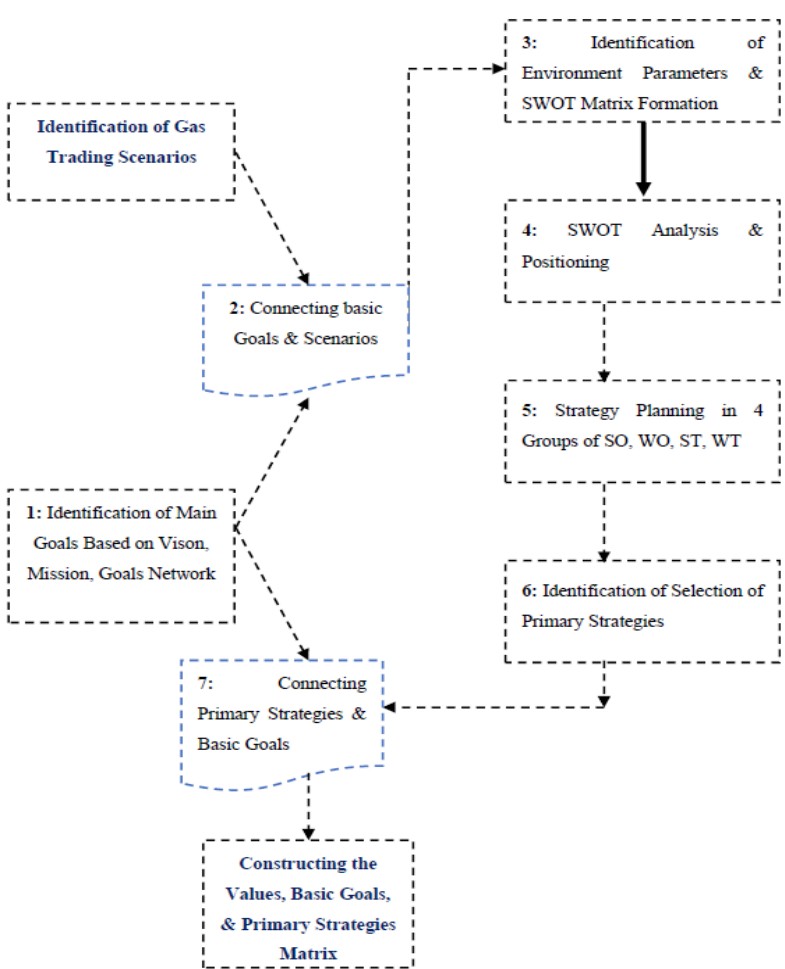

**Figure 4.** Making strategic decisions and developing strategies for NIGEC.

## 3.6. The Process of Goals Network Formulation

After determining the macro and functional strategies, the implementation of the strategic management, which is the last step after formulation and evaluation, is scheduled by performing the following procedures:

- Step 1: Determining the framework of decision's position.
- Step 2: Identifying the sets of goals.
- Step 3: Identifying the basic goals.
- Step 4: Checking out the ideal properties of the determined basic goals.
- Step 5: Highlighting the basic goals.
- Step 6: Developing the hierarchy of the basic goals as well as final-cross sectional goals network.

*3.7. Strategic Management: Control of Implementation*

The NIGEC uses the Balanced Score Card (BSC) method to control strategy implementation. BSC enables the organization to link short-term actions to the defined long-term strategy. For example, establishing relationships between balanced evaluations, annual budgeting, and planning enable the organization to determine its budget and to plan based on the defined strategy rather than on financial short-term goals (Kaplan and Norton 1992; Wu 2012; Hoque 2014; Rasoolimanesh et al. 2015; Strohhecker 2016).

However, BSC also enables the organization to adjust the remuneration system (rewards for managers and employees) based on the extent to which it has achieved the defined strategy and to avoid the use of short-term financial measures, like profit. Moreover, when BSC is employed as a strategic management system, managers will no longer be compelled to rely on short-term financial measures as the assessment measure in order to establish a link between practice and strategy. In this condition, four new processes are introduced, each of which, individually or in combination, link short-term actions with strategic long-term goals. Based on this method, after determining the quantitative cross-sectional goals, periodic monitoring measures are identified for every goal. The progress of the projects is assessed quarterly through the balanced scored card tools, and corrective actions on strategies, quantitative goals, and assessment measures are made, if necessary, by the technical team.

## 4. Conclusions

This study elaborates the conceptual framework designed for the strategic management department of a firm active in the oil industry of Iran. In the proposed methodology, a systemic approach is used to make sure the framework is not limited to any specific or predefined paradigm. Realistically, it attempts to focus on the specifics of the organisation, and then the characteristics of the environments to design the framework. It is argued in this study that the best approach is to develop the framework based on realizing the needs of the organisation, as there would be no right or better framework in strategic management. It is then of high significance to make sure the future is achievable with the designed framework. Based on this idea, the starting point is the visionary output, i.e., where the firm is aimed to be in its long-term plans, and accordingly, all other parts and steps are arranged for the organisation to reach that point. Using this idea enables the process of developing the conceptual framework to use all possible techniques and methods in the correct positions. Afterwards, by using network goals and value-focused thinking (VFT), the conceptual framework is implemented step-by-step, which is explained in detail through the study.

This study also explains the components of the framework and general rationale behind the steps taken to develop the framework in order to shed light on the processes required to develop a similar framework in another organisation. Regardless of the results of the implementation of the framework, which is beyond the aims of this study, it is argued that an adjusted conceptual framework can enable the managers of the organisation to react efficiently to the rapid altering environments, and cover all types of bottlenecks possible for a firm. Even though each conceptual framework for strategic management should be designed uniquely and based on the realities and requirements of the organisation, the proposed framework is aimed at bringing an insightful perspective for the development of a conceptual framework which, indeed, acts as the beating heart of the whole strategic management process.

It is discussed that the literature has neglected the prominent role of an exact and valid conceptual framework in the success of strategies implementation. Nonetheless, the conceptual framework enables the firms to create competencies because it clarifies the process of the strategic management for all units. It makes directing the organisation quite easier for the top managers, especially when unknown challenges are a fixed part of the environment. The best fit when the external environment is changing fast is never achieved unless the conceptual framework is adjusted with the fundamentals

of the modern organisations. Overall, this study is an attempt to prepare the organisation for future turbulence with a focus on the role and design know-how of the conceptual framework.

**Author Contributions:** Conceptualization, M.A. and E.M.; Data curation, M.A.; Formal analysis, M.A., E.M. and A.B.; Investigation, E.M. and A.B.; Methodology, M.A.; Project administration, M.A. and A.B.; Resources, M.A., E.M. and A.B.; Supervision, A.B.; Validation, E.M. and A.B.; Writing—original draft, M.A. and E.M.; Writing—review & editing, E.M. and A.B.

**Funding:** This research received no external funding.

**Conflicts of Interest:** The authors declare no conflict of interest.

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
