# Peer review of "Guidelines on Designing Conceptual Framework for Strategic Management with Application to the Energy Industry"

_admsci, doi:10.3390/admsci8030027_

Round 1

Reviewer 1 Report

I have attached the file for the review. 

Author Response

Dear Sir/Madam,

Thank you for your exact and helpful comments.

I have rechecked the paper for any grammar mistake.

Also, I have included some new chapters such as methodology in the paper to cover the missing parts.

I have enriched the conclusion.

The abstract is rewritten.

The obscure parts or irrelevant are omitted.

Quality of figures is improved.

Best Regards,

Reviewer 2 Report

I think this paper is very interesting as a operational guide, however it doesn't sound as a research paper to me.

Author Response

Dear Sir/Madam,

The paper is completey improved and new parts are added.

Best Regards

Reviewer 3 Report

General:
Overall, this paper presents an interesting perspective on strategic management frameworks within the context of an Iranian Oil company ( NIGEC). However, in its current form, the manuscript is not suitable for publication within the journal. I offer specific suggestions for improvement below.  

Specific comments:

1.  The introduction provides some arguments that sustain the case study like industry and country economic context, but there is room for improvements by adding some data that support the ideas, and moreover, relevant references form the international literature that debates the same topic.

2.  Being a case study, the paper should dedicate a separate part in the article to the NIGEC and explain why this case is relevant.

3.  The literature review part needs to contain also a part concerning models proposed by the literature, and then after the analyses of different models, the authors should introduce their model and underline elements from the literature, the novelty of the proposed model.

4.  The methodology is used not enough explained, please after section 2 Literature review insert a specific section where to describe the methodological aspects.

5.  Concerning the figures inserted in the paper, if there were not the author’s conception, it must be mentioned the reference.

6.  At page 6, first rows: Based on the framework of Mintzberg this model includes scenario planning thereby it can be interpreted in a descriptive paradigm. But, it also uses competitive and SWOT analyses hence it can also be included in the prescriptive paradigm (Van der Heijden 2011). More explanations are required to clarify the author contribution if it is this one.

7.  The third part Development of the Model is theoretical, containing a description of the so-called “model” elements based on the literature. Here author should add critical insights for each component related to the NIGEC case to support the model construction.

8.  The conclusions are underdeveloped; the author may highlight the paper's contribution to the literature. 

Author Response

Dear Sir/Madam,

Thank you for your helpful comments.

I have rechecked the paper for any grammar mistake.

Also, I have included some new chapters such as methodology in the paper to cover the missing parts.

I have enriched the conclusion.

The abstract is rewritten.

The obscure parts or irrelevant parts are omitted.

Quality of figures is improved.

The current paper is now more right to the point than the previous version.

Round 2

Reviewer 1 Report

Review of: Guidelines on Designing Conceptual Framework for Strategic Management with Application to Energy Industry

It is pleasing to see that the author(s) have improved the paper quite a lot. There are some additional suggestions below:

1)      It would be better to change the title of “Brief History of Strategic Management” to “Evolution of Strategic Management or Development of Strategic Management” as it would be more consistent with the main idea of the whole paper.

2)      At the end of the Section 2.1, it would be good to mention the current situation of the strategic management idea in the literature (maybe from 1990s until todays).

3)      What is the meaning of “in the firs” in the line 200?

4)      I insist you to explain the context of the study -i.e. NIGEC- in a few sentences since it is emphasised through the paper that the study mainly focuses on a specific organization.

5)      The quality of the Figure 3 must be improved (it is still blurry). Alternatively, it could be redrawn.

6)      Please recheck the numbering of the sections throughout the text.

7)      It would be good to rewrite the sentence in the line 387.

8)      Please paraphrased the last sentence of the conclusion.

9)      I suggest the author(s) to read the entire paper in order to check its grammatical accuracy as well as the typos.

Author Response

Dear Sir/Madam,

We have gone through the paper and have corrected the typos and other kinds of mistakes.

The figure is replaced with the original one.

A paragraph is added to the introduction to explain deatils of NIGEC.

The literature part about strategic management after 1980 is improved.

Best Regards

Reviewer 3 Report

General:
Overall, this paper presents an interesting perspective on strategic management frameworks within the context of an energy industry. There are some improvements, but not all the suggestions done were used by authors. I recall the specific suggestions for improvement below.  Or at least, the authors should explain why some suggestions were not explored in the improved version of the paper.
Specific comments:

1.  Being a case study, the paper should dedicate a separate part in the article to the NIGEC and explain why this case is relevant.

2.  In the section literature review there now there are included some references concerning models proposed by the literature, but then after the analyses of different models, the authors should underline elements from the literature, the novelty of the proposed model.

3.  The methodology is still not enough explained, please explain how starting from the literature review the framework was developed, what research methods and tools were employed. 

4.   The conclusions are underdeveloped; the author may highlight the paper's contribution to the literature and the connection with the literature used in the paper.

Author Response

(The authors gave the same response as above.)

Round 3

Reviewer 3 Report

Overall, this paper presents an interesting perspective on strategic management frameworks within the context of an energy industry. There are few improvements, but not enough. I recall the specific suggestions for improvement below.  Or at least, the authors should explain why some suggestions were not explored in the improved version of the paper.
Specific comments:

1.  In the section literature review there now there are included some references concerning models proposed by the literature, but then after the analyses of different models, the authors should underline elements from the literature, the novelty of the proposed model.

2.  The methodology is still not enough explained, please explain how starting from the literature review the framework was developed, what research methods and tools were employed. 

3.   The conclusions are underdeveloped; the author may highlight the paper's contribution to the literature and the connection with the literature used in the paper.

Author Response

Dear Sir/Madam,

As it is explained in the 2.5, conceptual frameworks presented in the litearture are not really elaborated except in few cases making it very hard to describe how much innovation our proposed model has brought. But the main innovation of this model is developing the frameowrk with a foresight approach, starting from the future to be. This is now expalined with mor details in its part.

The conclusion and methodology are also explained with more deatils.

Best Regards,